# How does Sec63 affect the conformation of Sec61 in yeast?

**Pratiti Bhadra[1], Lalitha Yadhanapudi[2], Karin Römisch[2], Volkhard Helms[1]***

**1** Center for Bioinformatics, Saarland University, Saarbrücken, Saarland, Germany, **2** Faculty of Natural Sciences and Technology, Saarland University, Saarbrücken, Saarland, Germany

* volkhard.helms@bioinformatik.uni-saarland.de

## Abstract

The Sec complex catalyzes the translocation of proteins of the secretory pathway into the endoplasmic reticulum and the integration of membrane proteins into the endoplasmic reticulum membrane. Some substrate peptides require the presence and involvement of accessory proteins such as Sec63. Recently, a structure of the Sec complex from *Saccharomyces cerevisiae*, consisting of the Sec61 channel and the Sec62, Sec63, Sec71 and Sec72 proteins was determined by cryo-electron microscopy (cryo-EM). Here, we show by co-precipitation that the Sec61 channel subunit Sbh1 is not required for formation of stable Sec63-Sec61 contacts. Molecular dynamics simulations started from the cryo-EM conformation of Sec61 bound to Sec63 and of unbound Sec61 revealed how Sec63 affects the conformation of Sec61 lateral gate, plug, pore region and pore ring diameter via three intermolecular contact regions. Molecular docking of SRP-dependent vs. SRP-independent signal peptide chains into the Sec61 channel showed that the pore regions affected by presence/absence of Sec63 play a crucial role in positioning the signal anchors of SRP-dependent substrates nearby the lateral gate.

**Data Availability Statement:** The simulation methods are provided in sufficient detail in the method section. All MD trajectory files will be provided by the corresponding author upon request (no restrictions). All analysis scripts are

## Author summary

The ribosome particles of a cell constantly synthesize fresh proteins. Some of these will be either integrated into organellar membranes or will be secreted from the cell. To this aim, they need to enter a channel protein complex termed translocon in the membrane of the endoplasmatic reticulum. The translocon needs to determine based on certain sequence features whether the peptide should be translocated into the endoplasmatic reticulum lumen or whether opening of a lateral gate will enable that its transmembrane domains slide laterally into the membrane. Whereas it is known that some peptides cannot be translocated without the involvement of accessory membrane proteins such as the Sec63 protein, the mechanistic details how Sec63 affects the conformation and/or dynamics of the translocon are so far not known. Here, we used molecular simulations, molecular docking and co-precipitation experiments to characterize protein contact residues and conformational shifts in the channel pore. Our simulations reveal that the plug moiety adopts different conformations in the channel in the presence, respectively the absence of Sec63, and we describe the influence of Sec63 on the conformation of the lateral gate. This

available at https://github.com/pratitibhadra/
Sec63_Sec61.git.

**Funding:** VH received funding by Deutsche
Forschungsgemeinschaft via grant He3875/15-1.
The funders had no role in study design, data
collection and analysis, decision to publish, or
preparation of the manuscript.

**Competing interests:** The authors have declared
that no competing interests exist.

study contributes to our understanding of the functionality of translocon and accessory proteins, and may eventually aid in characterizing and overcoming the effects of mutations related to protein translocation defects.

## Introduction

In eukaryotes, newly synthesized polypeptide chains destined for the secretory pathway are translocated into the endoplasmic reticulum (ER) or laterally inserted into the ER membrane via a conserved hetero-trimeric complex, the Sec61 channel [1]. In prokaryotes, the universally conserved protein conducting channel is known as the SecY complex. Successful protein translocation is essential for the proper functioning of cells. Defects in protein translocation have been linked to many diseases, including cancer and hereditary human diseases [2]. Translocation of secretory proteins can occur either co-translationally or in a post-translational manner [2–4]. In the co-translational pathway, a newly synthesized protein bearing a signal sequence exits from the ribosome and a ribonucleoprotein particle called signal recognition particle (SRP) binds to the signal sequence. Subsequently, upon interaction of SRP with the SRP receptor at the ER membrane, SRP dissociates from the signal sequence, which is then free to insert into the Sec61 channel. In the post-translational pathway, the fully synthesized secretory precursor or protein is released from the ribosome and then, supported by chaperones and/or SRP-independent machineries, enters into the Sec61 channel. Recently published cryo-electron microscopy structures [5, 6] of the heptameric Sec complex that mediates post-translational translocation into the yeast ER provide mechanistic insight into the post-translational process [5–8]. In yeast cells, the Sec complex is composed of two protein complexes, the Sec61 complex and the hetero-tetrameric Sec62-Sec63 complex comprising Sec62, Sec63, Sec71 and Sec72. Sec62 and Sec63 are also present in mammalian cells [9].

In *Saccharomyces cerevisiae*, the Sec61 complex/channel is composed of the three subunits, Sec61 (the pore-forming subunit), Sbh1, and Sss1. In higher eukaryotes, Sbh1 and Sss1 are known as Sec61$\beta$ and Sec61$\gamma$. Sec61 forms a compact helix bundle of 10 transmembrane (TM) helices that can open the channel pore across the membrane. The pore can also open sideways towards the lipid bilayer via a lateral gate, which is formed by a pair of TM helices, TM2 and TM7. The channel pore further exhibits a pore ring of six hydrophobic amino acids residues [10] and an $\alpha$-helical plug moiety that rest inside the pore on the lumenal side to occlude the channel [6]. In yeast, the Sec61 complex and Sec63 are involved in co- and post-translational protein translocation [11]. Cross-linking data in mammalian cells suggested that interaction of ribosomes and Sec62 with Sec61 is mutually exclusive and that the SRP receptor plays a specific role in the interconversion of translocons from Sec62-dependent to SRP-dependent translocation [12].

Sec63 contains three TM segments, a large cytosolic Brl domain and a J domain between the second and third TMs. The Brl domain of Sec63 makes contact with the cytosolic loops TM6-TM7 and TM8-TM9 of Sec61. The TMs of Sec63 interact with TM1 of Sec61, and the TM of Sbh1 and Sss1 as well. The structure of Sec62 and the J domain of Sec63, which interacts with BiP [13], and C-terminus of Sec63 were not resolved in the cryo-EM structures of the Sec complex. Recently, Yim *et al.* [14] suggested that Sec62, Sec63, Sec71 and Sec72 dynamically associate with the Sec61 channel for binding of signal sequences and the initiation of protein translocation. In addition, a recent study suggested a role for the Sec62-Sec63 complex in membrane protein insertion and topogenesis of signal anchor proteins [15, 16]. Earlier, Ng *et al.* [17] had shown that mutant alleles of *SEC62* and *SEC63* impaired post-translational

translocation of SRP-independent proteins *in vivo* and *in vitro*, whereas co-translational translocation of SRP-dependent proteins was unaffected in yeast, suggesting that the mode of translocation is determined by the signal sequence.

Structural studies indicate that Sec61/Sec61$\alpha$ undergo considerable conformational changes during translocation both in the yeast and the mammalian complex [18, 19]. Recent cryo-EM structures of the Sec complex from yeast demonstrated that binding of the Sec62-Sec63 complex to the Sec61 complex in the post-translational condition caused a wide opening of the lateral gate [5, 6]. In comparison, ribosome binding opens the lateral gate to a lesser extent, and even after insertion of a nascent polypeptide chain, the ribosome-associated Sec61 channel is not as widely open as the Sec61 channel in the heptameric Sec complex [6, 7].

Over the last few years, molecular dynamics (MD) studies have focused on characterizing mechanistic structural and energetic details of how the Sec61 channel recognizes substrate peptides and directs them either to membrane insertion or translocation [20–23]. Using a microsomal *in vitro* expression system and MD simulations, Jaud *et al.* examined the Sec61 insertion efficiency of short polyleucine segments [24]. Combination of mutagenesis and MD simulation revealed that membrane insertion strongly depends not only on hydrophobicity but also on the local conditions of the Sec61 translocon [25]. These results suggested that hydrophobic mismatch controls the insertion across the ER membrane. Using coarse-grained MD simulations, Sun *et al.* showed that the lateral gate of the mammalian Sec61$\alpha$ can recover the partially-closed state rapidly after a transmembrane segment enters the bilayer [26]. They suggested that the plug movement and the pore ring conformation are linked to the closing mechanism of the lateral gate. Inclusion of a hydrophobic peptide substrate in the translocon was reported to stabilize the opening of the lateral gate for membrane integration, whereas a hydrophilic peptide substrate favored the closed lateral gate conformation [27]. Also, CG-(coarse-grained)-MD simulations were performed to study how nascent chains insert into the membrane or get translocated [28]. A minute-timescale CG simulation provided insight into the mechanisms, kinetics, and regulation of Sec61 channel-facilitated protein translocation and membrane integration [29]. A refined CG model [30] yielded quantitative agreement with experimental data in terms of membrane integration of the hydrophobic segment and translocation of Type 1 ($N_{Lumen}$) and Type 2 ($N_{Cytosol}$) signal sequences, respectively. None of these previous MD studies has, however, addressed how conformational and dynamic aspects of the central Sec61 channel are affected by its interaction with the Sec62-Sec63 complex. Do the accessory proteins regulate the conformation of the pore ring, the plug, and the lateral gate? Whereas TM2 and TM7 helices of the Sec61 channel have been identified as the signal sequence binding site using a limited number of model proteins [19, 31, 32], the interaction of diverse signal sequences with the lateral gate remains unclear.

In this study, we thus characterized the role of the Sec63 protein on the conformational dynamics of the yeast Sec61 translocon using atomistic molecular dynamics simulations. Our simulation results reveal that the open pore is caused by a reorientation of TM4 of Sec61 which is governed by the interaction between TM3 of Sec63 and TM1 of Sec61. We demonstrate that the plug moiety adopts different conformations in the channel in the presence, respectively the absence of Sec63, and describe the influence of Sec63 on the conformation of the lateral gate. Additionally, using co-precipitation experiments, we demonstrate that Sbh1 is not an essential component for the Sec complex stability. Furthermore, using molecular docking, we shed light on the interaction of different signal sequences with the lateral gate of Sec61 in its conformation when bound to the Sec62-Sec63 complex in the Sec complex. We observed that the lateral-gate region of Sec61 formed tighter interactions with the hydrophobic core of SRP-dependent (signal anchor) substrates than of SRP-independent (signal sequence) substrates. This confirms the importance of signal anchor hydrophobicity as noticed before.

## Materials and methods

### Homology modeling

The cryo-EM structure of the yeast Sec complex (PDB ID: 6N3Q; resolution: 3.68 Å) provides structural information on the assembly of Sec61, Sss1, Sbh1 and Sec63, although not all domains were resolved/detected [5]. The structural information of some missing parts of 6N3Q was modeled using another cryo-EM structure, 6ND1 [6] (resolution: 4.10 Å). Long stretches of 25 and 36 amino-acid residues from the N-termini of Sss1 and Sbh1 are missing in both cryo-EM structures, respectively. Similarly, structural information of the J-domain (86 amino acids long) and 63 amino-acid residues from the C-terminus of Sec63 are also missing. We did not model these long missing parts of Sss1, Sbh1 and Sec63. A shorter than 10 amino acids missing region (in both cryo-EM structures: D79-S86 and I551-T556) of the Sec63 protein was modeled as loop conformation. The structural information of the missing parts (missing in both structures 6N3Q and 6ND1) of the Sec61 subunit was obtained by homology modeling based on the mammalian Sec61α subunit (PDB 2WWB, 6.48 Å) [33] as mammalian Sec61α and yeast Sec61 share 56.6% sequence identity. Homology modeling was performed using MODELLER9.21 [34]. This assembly was used for MD simulations of the Sec61 complex with Sec63, and will subsequently be addressed as 'Sec63-bound' state. We simply removed Sec63 for the simulations of the Sec61 complex without Sec63, which we will term 'free' state in the following.

### Molecular dynamics (MD) simulation

The above described conformations of the yeast Sec61 complex (Sec61, Sss1 and Sbh1) without and with Sec63 protein were used as the starting structures for our MD simulations of free and Sec63-bound states, respectively. To mimic the Sec61 complex in yeast cells, the starting methionine of Sec61 was cleaved off and Ser2 was N-acetylated [35, 36]. The protonation states of titratable amino acids were determined using the web server PDB2PQR [37]. The protein models were placed in a POPC bilayer membrane using CHARMM-GUI [38, 39]. The orientation of the proteins in the membrane was determined applying the PPM server [40], 150 mM of physiological KCl salt condition were used; the overall system was electrically neutral. The unbound state system consisted of 148,218 atoms including 300 POPC molecules, 32,933 water molecules, 89 potassium ions, and 104 chloride ions. The bound state system consisted of 244,258 atoms including 440 POPC molecules, 55,948 water molecules, 152 potassium ions, and 165 chloride ions. The missing regions of Sec63 split the Sec63 protein into three intermittent parts. These parts were treated as different chains in the MD simulations. Simulations were performed using the GROMACS2018.8 software package [41]. The protein groups, lipids and water were represented in atomistic detail using the CHARMM36 force field for lipids [42] and proteins [43] and the TIP3P [44] model for water. The starting structures of the two systems are shown in Fig 1.

Initially, only water molecules and ions were energy minimized by steepest descent energy minimization (5000 steps) and equilibrated in the NVT ensemble (303.15K) while membrane lipids and protein atoms were kept fixed. This was followed by further minimization of the whole system whereby the positions of protein backbone atoms were harmonically restrained and all other atoms could move freely. The minimization was followed by 400 ps of MD equilibration as prescribed by the CHARMM-GUI server [39]. A six-step equilibration procedure was applied, which included harmonic restraints on heavy atoms of the protein, planar restraints to hold the position of lipid head groups of the membrane along the Z-axis, and dihedral restraints to keep fatty acid chain double bonds in the cis conformation. The systems

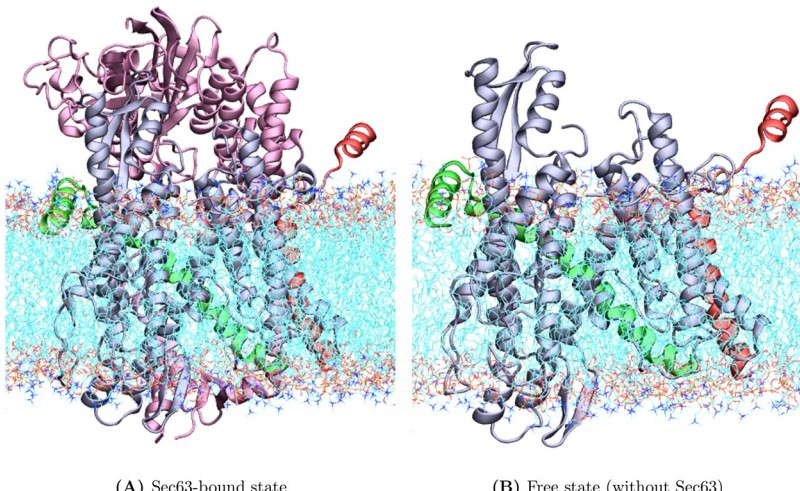

**(A)** Sec63-bound state          **(B)** Free state (without Sec63)

**Fig 1. Initial conformations of the molecular dynamics simulations for (A) Sec63-bound state; and (B) free state without Sec63.** Sec61, Sss1, Sbh1 and Sec63 are shown in ice-blue, green, red and mauve, respectively, with "New Cartoon" model. The regions populated by lipid molecules are marked using the "line" method. Water and ion molecules are omitted for clarity. The image was generated using VMD-1.9.4. [45].

were relaxed gradually by applying decreasing force constants during the equilibration that ranged from 4000 to 50 kJ/mol/nm$^2$ on heavy atoms of the protein backbones, from 2000 to 0 kJ/mol/nm$^2$ on heavy atoms of protein side chains, and from 1000 to 0 kJ/mol/nm$^2$ on lipid molecules. Finally, 1 $\mu$s long simulations were performed in the NPT ensemble at the experimental temperature of 303.15K without applying any restraints. Periodic boundary conditions in all three dimensions were used. The particle-based cut-off method (Verlet) was employed for neighbor searching. Short-ranged Coulomb interactions were computed using a cut-off of 1.2 nm. Long-range electrostatics were handled by the particle-mesh Ewald (PME) method [46]. The nonbonded Lennard-Jones interactions were calculated using a 1.2 nm cut-off with a smooth switching function beginning at 1 nm. The Nose-Hoover thermostat [47] was used to keep the temperature constant and an atmospheric pressure of 1 bar was applied using the Parrinello-Rahman barostat [48]. To collect representative statistics, we performed five independent simulations for each state with different starting velocities.

## Conformational analysis

The effects of Sec63 on the global structural plasticity of Sec61 were estimated in terms of RMSD, Rg and RMSF. Distance and angle based measures were employed to investigate the local dynamics of Sec61. The end point of a transmembrane (TM) helix was defined by the center of mass of the C$\alpha$ atoms of the four terminal residues (one helical turn). The helical axis of a TM helix connects its two end points. The secondary structures were assigned with the Dictionary of Secondary Structure of Proteins (DSSP) algorithm [49].

The RMSD (Fig 2) and the radius of gyration (Fig 3) have reached plateau values after about 600 ns suggesting that the conformational relaxation of the unbound simulation has taken place by that time. We may assume that this also applies to local structural features such as the distance and angle measurements of the pore-ring, lateral gate and plug. S1, S2 and S3 Figs show that the distributions obtained from the last 50 ns are narrower than for the last 400 ns. Yet, the peak positions of the average distributions (grey color) are located at similar values in both states ('free' and 'Sec63-bound') for both time-windows (last 400 ns and last 50 ns).

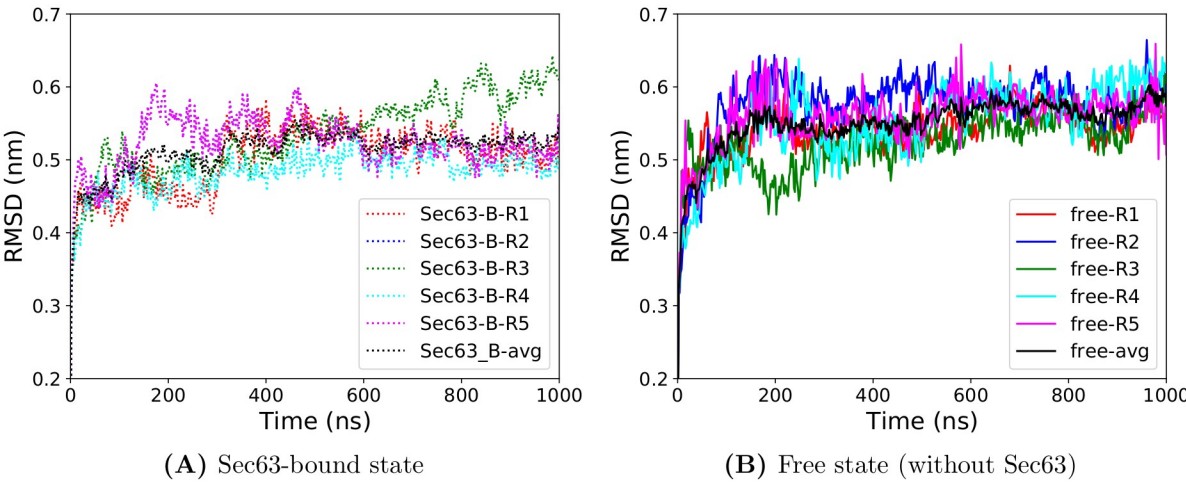

**(A)** Sec63-bound state                **(B)** Free state (without Sec63)

**Fig 2. RMSD values of Sec61 with respect to the initial starting structure along with simulation time obtained from five independent MD simulations (Ri, i = replica number).** Here and in other figures, 'Sec63-B' and 'free' denote 'Sec63-bound' and 'free' state, respectively. (A) Sec63-bound state (B) Free state (without Sec63).

Comparing the statistical distributions of these three measures of local dynamics suggests that the last 50 ns of the trajectories equally well describe the local dynamics as the last 400 ns of the trajectories. Therefore, in the remainder of the manuscript we used the last 50 ns of the 1 $\mu$s long MD simulations to analyze the local flexibility of Sec61. GROMACS utilities and in-house scripts were used for analysis. Statistical $p$-values were calculated using the hypergeometric test.

## Docking method

Previous experimental studies showed varying degrees of translocation defects of SRP-independent and SRP-dependent substrates in SEC62 mutant, SEC63 mutant and SEC72 deletion strains [14, 15, 17, 50]. The hydrophobic cores of the signal sequences of SRP-dependent

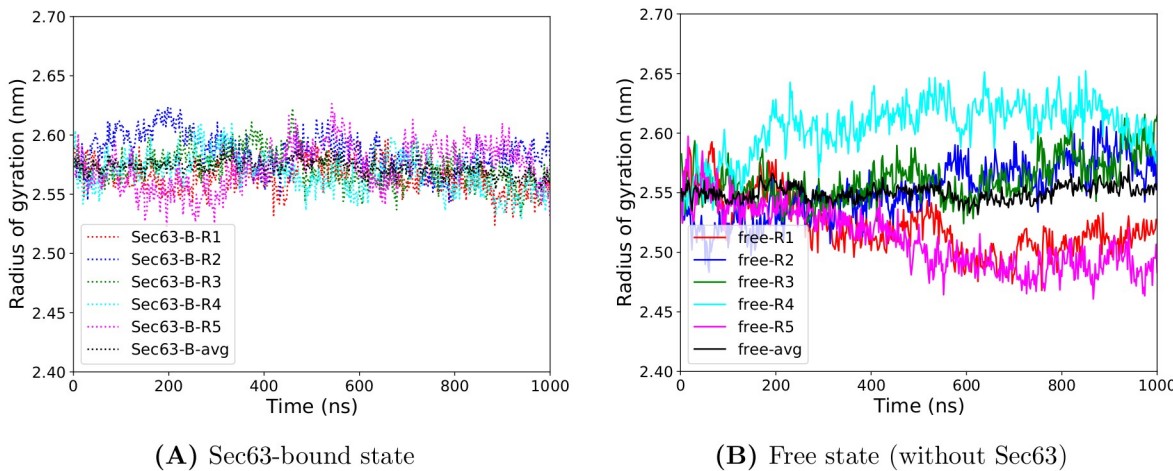

**(A)** Sec63-bound state                **(B)** Free state (without Sec63)

**Fig 3. Time dependence of the radius of gyration ($R_g$) of Sec61 obtained from five independent MD simulations (Ri, i = replica number).** (A) Sec63-bound state (B) Free state (without Sec63).

substrates possess a substantially greater average hydrophobicity than SRP-independent substrates [17, 51]. Therefore, the aim of the docking part was to investigate the interaction of the hydrophobic cores of signal sequences with Sec61 in its conformation when bound to the Sec62-Sec63 complex (Sec62/Sec63/Sec71/Sec72), which is believed to reflect its conformation during post-translational protein translocation. We selected the hydrophobic cores of the signal sequences of the Sec61 substrates CPY, Gas1, PDI, pp$\alpha$-factor, Dap2 and Pho8 (see S1 Table) for our study. CPY, Gas1, PDI and pp$\alpha$-factor are SRP-independent substrates [17] with N'-terminal cleavable signal sequences. Dap2 and Pho8 are SRP-dependent substrates [17] and contain a noncleavable transmembrane signal sequence (signal-anchor sequence). As the signal-anchor sequences of Dap2 and Pho8 are longer than 50 amino acids docking of the full signal sequences is not feasible due to limitations of the AutoDock CrankPep (ADCP) software posed on the maximal length of the peptide [52].

The homology-based structural model of Sec61 described above was used for docking with ADCP. ADCP makes use of CRANKITE [53], a software package that samples the conformational space of proteins using a Metropolis Monte Carlo method. In ADCP, CRANKITE is combined with the grid-based AutoDock representation of a rigid receptor to optimize peptide conformations and peptide-receptor interactions at the same time. Thus, we adopted a relatively large grid box of dimensions 90 Å X 90 Å X 126 Å, centered at the center of mass of Sec61 to cover the channel pore. Three independent sets of docking runs were conducted that were started with different random seeds. All clustered pockets within the box with energy below 0 kcal/mol were considered as potential binding pockets. The number of replicas for each docking run was set to 100 with $1.75 \times 10^{8}$ evaluations of the scoring function for each replica. The partition parameter was set to 20 meaning that 20% of the replicas were started with the peptide sequence modeled initially into a helical conformations and the remaining 80% were started from an extended conformation. All other docking parameters were kept at the default values of ADCP. Each independent docking run provided top ten final docking poses with ranking after clustering. The top ten solutions from three independent docking runs (N = 30) were considered for analyzing the interaction between signal sequences and Sec61. The minimum distance between heavy atoms of two residues was considered as residue-residue distance. S4 Fig compares docking solutions of the full-length signal sequence of CPY to docking solutions for its hydrophobic core. For 18 out of 30 docking poses, structural matches below 2.0 Å RMSD were obtained. Pose #2 has the lowest RMSD of 0.76 Å from pose #5 of the hydrophobic core. At least for the relatively short CPY signal sequence, this exercise showed that docking of the hydrophobic core identified poses that are representative of the full length signal sequence.

## Coprecipitation of Sec63 and Sec61 complexes with Concanavalin A-Sepharose

The strain H3232 (MATa KanMx::sbh1 HphMx::sbh2 leu2-3,112 ura3-52 GAL$^{+}$) [54] was grown to early exponential phase at 24 ˚C or shifted to 38 ˚C for 3h prior to spheroplasting and microsome preparation as described in Pilon *et al.* study [55]. Membranes were lysed in solubilization buffer containing 50 mM HEPES-KOH pH 7.4, 400 mM KAc, 5 mM MgAc, 10% (w/v) glycerol, 0.05% (v/v) $\beta$-mercaptoethanol, 1X Protease Inhibitor Cocktail (Roche), and 3% (w/v) digitonin or 1% (w/v) Triton X-100. Complexes were precipitated with Concanavalin A, which was equilibrated with 50 mM HEPES-KOH pH 7.4, 10% (w/v) glycerol, 0.05% (v/v) $\beta$-mercaptoethanol, 1X Protease Inhibitor Cocktail (Roche), and 1% (w/v) digitonin or 1% (w/v) Triton X-100 as described in [55]. Proteins in the supernatants were precipitated with 20% (w/v) trichloro acetic acid prior to solubilization in SDS-PAGE buffer for 10

min at 65 ˚C. Proteins bound to Concanavalin A-Sepharose were denatured by heating the beads in SDS-PAGE buffer to 65 ˚C for 10 min. Equivalent amounts of bound and free complexes were loaded onto 10% Bolt Bis-Tris gels (InVitrogen). Proteins were transferred to nitrocellulose and Sec63, Sec61, and Sss1 were detected using specific antibodies against the Sec63 J-domain, the Sec61 C-terminus, or Sss1. Antibodies against Sec63 and Sss1 were gifts from Randy Schekman.

## Results

### Structural stability of channel pore and interaction sites between Sec61 and Sec63

In the initial analysis, we inspect whether the starting conformation (full-length Sec61 and partial-length Sbh1, Sss1, and Sec63) was stable during the MD simulations. RMSD values calculated relative to the starting structure for the backbone atoms of all Sec61 residues (Fig 2) were slightly lower (0.53 nm) in the Sec63-bound conformations (average over 5 independent simulations considering the last 100 ns trajectory) than for the free state conformations (0.58 nm). This difference is statistically significant (hypergeometric $p$-value = 0.02, with threshold 0.55 nm). Fig 4 shows the RMSD alignment of the final conformations (at 1$\mu s$). At the same time, there is no large difference in the average radius of gyration values of Sec61 in 'Sec63-bound', as well as the 'free' states. In two independent free simulations (see Fig 3), the radius of gyration (Rg) of Sec61 showed a drop in the value after 500 ns and a larger deviation among the unbound simulations, suggesting that Sec63 has a stabilizing role on Sec61.

According to the cryo-EM structure, Sec63 makes contact with Sec61 through its cytosolic, transmembrane, and luminal domains (Fig 5A). In the cytosolic region (interaction site 1), the Fibronectin type III (FN3) domain of Sec63 (mauve) interacts with the cytosolic loop between TM6 and TM7 of Sec61 (gray) [5]. In the membrane (interaction site 2 and 4), TM3 of Sec63 (mauve) is located close to the highly conserved TM of Sbh1 (red) and TM1 of Sec61 (gray). In the ER lumen (interaction site 3), the C-terminus of the J-domain of Sec63 (mauve) interacts as $\beta$-sheet with the lumenal loop between TM5 and TM6 of Sec61 (gray). Distances between

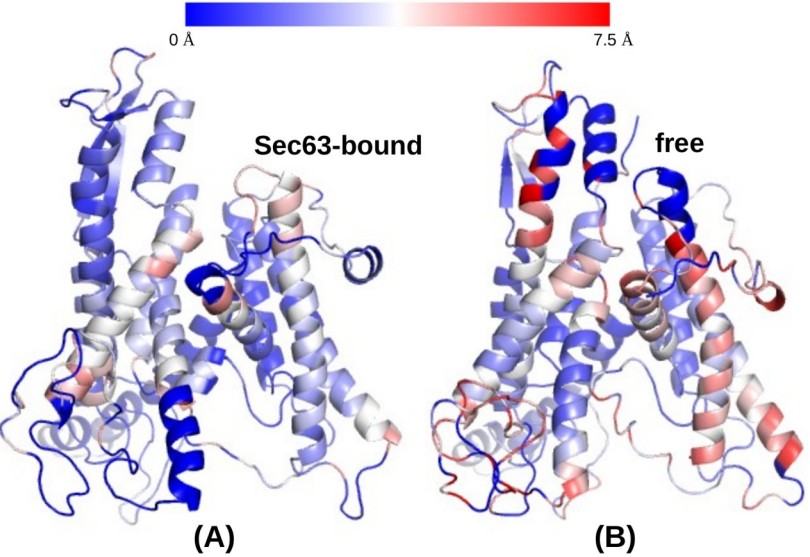

**Fig 4. RMSD between the final conformations (at 1 $\mu$s) of Sec61 of (A) Sec63-bound state and (B) free state, and the starting modeled structures shown in Fig 1.** Smaller RMSD deviations are represented by blue color and larger deviations are colored red, see color bar on top.

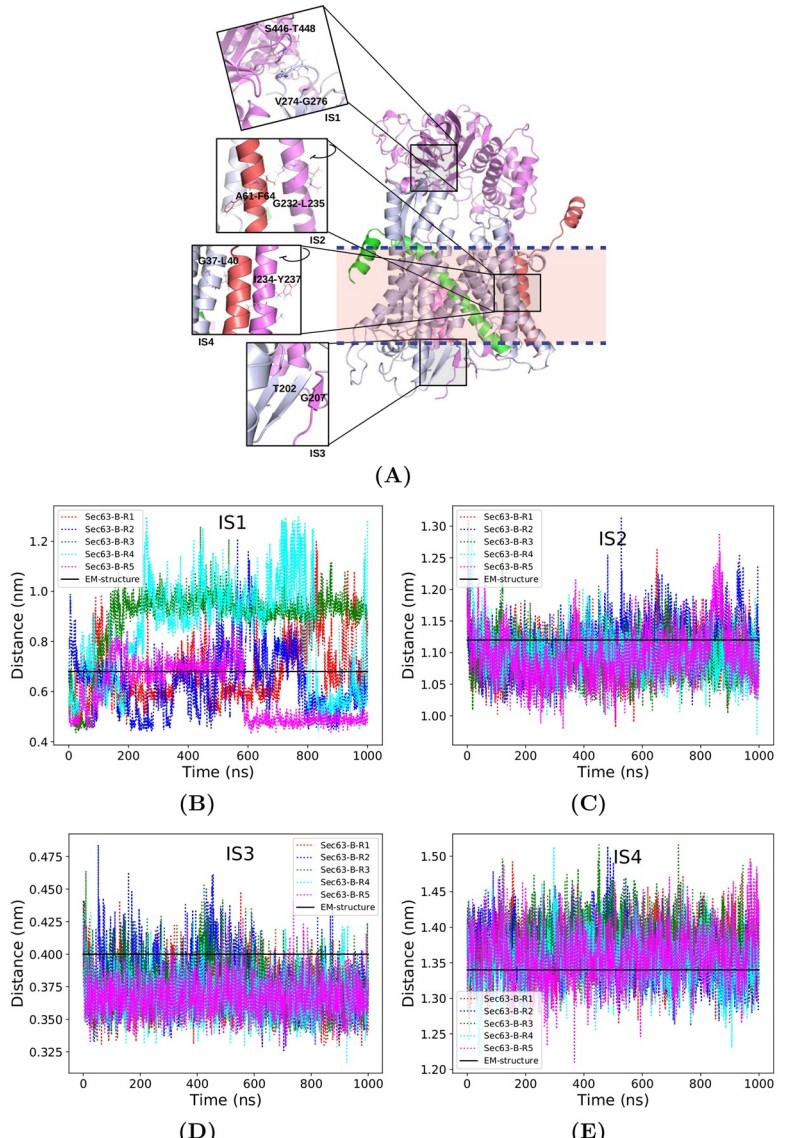

**Fig 5. Interaction sites of Sec63 (mauve) and Sec61 complex.** Sec61, Sbh1 and Sss1 are colored gray, red, and green, respectively. The light pink rectangle box with blue dashed borders represents the lipid bilayer. (A) Magnified views of the interaction sites. The evolution of the contact distances at interaction sites along with simulation time (B) interaction site 1 (IS1—V274-G276:Sec61 and S446-T448:Sec63), (C) interaction site 2 (IS2—A61-F63:Sbh1 and G232-L235:Sec63), (D) interaction site 3 (IS3—T202:Sec61 and G207:Sec63) and (E) interaction site 4 (IS4—G37-L40: Sec61 and I234-Y237:Sec63) during five independent MD simulations (Ri, i = replica number). The black horizontal line denotes the respective value in the cryo-EM structure.

the two proteins at the interaction sites were monitored as a function of simulation time to check the stability of the contacts (Fig 5B, 5C, 5D and 5E). The closest residue pairs from both proteins at the interaction sites were considered as interacting residues. The contact distance at interaction site 1 (IS1) was defined as the distance between the center of mass (COM) of the $C\alpha$ atoms of the regions V274-G276:Sec61 and S446-T448:Sec63. Similarly, the contact distances at interaction site 2 (IS2) and interaction site 4 (IS4) were measured between A61-F63: Sbh1 and G232-L235:Sec63 (IS2) and G37-L40:Sec61 and I234-Y237:Sec63, respectively. The

distance between the COM of residues T202:Sec61 and G207:Sec63 was considered as the contact distance at interaction site 3 (IS3). Fig 5C, 5D and 5E show that the contacts at IS2, IS3 and IS4 were well maintained, likely due to the regular secondary structure at those regions even though only part of Sec63 was present in the MD simulations. In contrast, Fig 5B illustrates that the contact distance at IS1 fluctuated throughout the trajectories. Thus, this contact was not stable. It is possible that these unexpected fluctuations at IS1 are related to the missing Sec63 C-terminus, which is located near the IS1 region. IS1 is situated in a loop region in both Sec61 and Sec63. Besides, it is located almost $\sim 3.0$ nm away from the phosphate head group of the lipid bilayer, suggesting that the full solvent exposure of this region contributes to its increased flexibility.

## Conformational changes in Sec61 at interaction sites with Sec63

To examine the residue-specific flexibility, root mean square fluctuations (RMSF) were calculated for each residue from the simulation trajectories. The RMSF plot (average over five independent simulations, S5(C) Fig) indicates that the regions surrounding IS1 and IS3 in Sec61 were slightly more flexible in the free state compared to the Sec63-bound state. There were no significant differences in other regions of Sec61. To further characterize the conformational changes in Sec61 at IS3 (P200-E212) and IS1 (R264-P282), we computed DSSP [49] profiles of these regions (S6 and S7 Figs) using the final 50 ns from the trajectories. In the absence of Sec63, both interaction sites gradually lost the regular $\beta$-sheet conformation with increasing simulation time. To quantify this, the probability distributions of the percentage of residues of IS3 belonging to $\beta$-sheet (Fig 6) exhibit that the $\beta$-sheet structure was transformed to an irregular structure in three free (without Sec63) simulations, whereas it was well maintained in all Sec63-bound state simulations (hypergeometric $p$-value = 0.08, threshold: the percentage of residues adopting $\beta$-sheet in the simulation < 20%). A similar tendency was also observed for IS1 (S8 Fig). The simulation results suggest a partial unfolding of IS3 in the absence of Sec63 in agreement with the available cryo-EM structures. In contrast to the IS3 region, the secondary

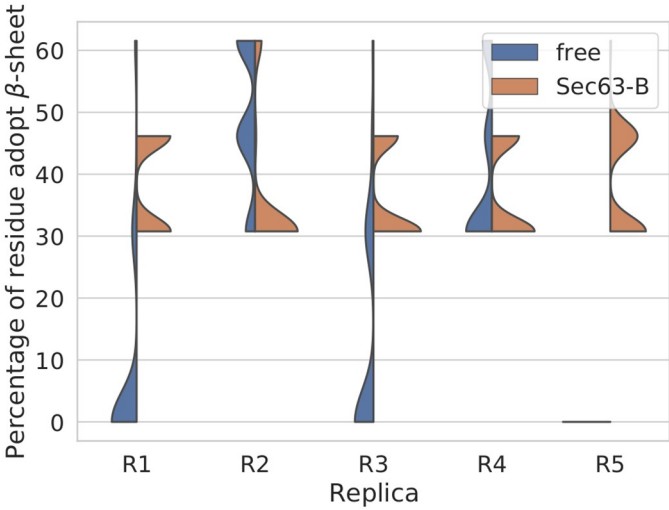

**Fig 6. Probability distribution of the percentage of residues of Sec61 at IS3 (P200-E212) adopting a $\beta$-sheet conformation during the last 50 ns of the MD simulations.** Sec63-B and free denote 'Sec63-bound' and 'free' states, respectively. The percentage values are 46% and 0% in the cryo-EM structures of Sec complex and ribosome-Sec61 complex, respectively.

structure of the IS1 region in the free state does not match with its conformation in the cryo-EM structure of the ribosome-Sec61 complex. The cytosolic loop between TM6 and TM7 of Sec61 probably adopts a $\beta$-sheet conformation in the presence of a binding partner regardless of the partner. The superimposed conformations of the equillibriated representative of Sec61 at Sec63-bound and free states (at 1 $\mu$s) demonstrate the conformational changes at the interaction regions, see S9 Fig.

## The Sbh1 transmembrane helix is not critical for Sec complex stability

In the cytosolic (IS1) and luminal (IS3) interaction sites, Sec63 strongly interacts with only Sec61. TM3 of Sec63, however, makes contact with both Sec61 and Sbh1 at the membrane interaction site (IS4 and IS2) (Fig 5). Sec63-bound state simulations show that the contact between TM of Sbh1 and TM3 of Sec63 is stable (see Fig 5E). To investigate whether the interaction between Sbh1 and Sec63 (IS2) contributes to the stability of contacts between the Sec61 complex and Sec63, we analyzed Sec complex stability using co-precipitation experiments.

Simultaneous deletion of SBH1 and its homolog SBH2 leads to temperature-sensitive growth of the resulting Δsbh1 Δsbh2 yeast strain, likely due to a translocation defect for an essential secretory protein [54]. Temperature-sensitivity of the strain can be rescued by expressing a truncated SBH1 encoding only the transmembrane helix; this implied an essential function for the Sbh1 transmembrane domain [54]. The Sec complex structure (Fig. 3 of [54]) suggests that this function may be the stabilization of the Sec63-Sec61 complex interaction.

We therefore asked whether the contact between TM3 of Sec63 and the Sbh1 transmembrane helix at IS2 was critical for Sec complex stability. We grew a Δsbh1 Δsbh2 strain either at permissive temperature for ER import (24 °C) or shifted the cells to restrictive temperature (38 °C) for 3h prior to preparation of microsomes. We solubilized microsomes from both growth conditions in 3% (w/v) digitonin which preserves Sec complex integrity [55]. After sedimenting insoluble material by centrifugation at 13,200 x g, we precipitated the Sec63 complex from the supernatants via its only glycan moiety on Sec71 using Concanavalin A, a lectin that binds the mannose residues in N- and O-linked glycans [56]. Since the Sec61 complex does not contain glycosylated subunits, it only coprecipitates with ConcanavalinA-Sepharose when it is associated with the Sec63 complex. As a control we used membranes lysed in 1% (w/v) Triton X-100 which dissociates the Sec61 complex from the Sec63 complex [55].

As shown in Fig 7, in lysates derived from Δsbh1 Δsbh2 yeast grown at 24 °C Sec61 coprecipitated with the Sec63 complex in digitonin, but not in Triton X-100 (Fig 7 left, lanes 3 and 4 vs. lanes 1 and 2). In addition, we blotted for the small Sec61 complex subunit Sss1 which also coprecipitated with the Sec63 complex and hence bound to Concanavalin A-Sepharose in digitonin lysates, but not in Triton X-100 (Fig 7 left, lanes 3 and 4 vs. lanes 1 and 2). In lysates derived from Δsbh1 Δsbh2 yeast shifted to the restrictive temperature (38 °C) for 3h, Sec61 and Sss1 were also found bound to the Sec63 complex in digitonin, but not in Triton X-100 (Fig 7 right, lanes 7 and 8 vs. lanes 5 and 6). We therefore conclude that the Sec63 contact to the Sbh1 transmembrane helix at IS2 does not contribute critically to Sec complex stability.

## Conformational changes at the pore-ring, the plug and the lateral-gate of Sec61

The Sec61 channel exhibits several structural elements critical for protein translocation, namely a lateral gate, a pore-ring of hydrophobic amino acid residues, as well as an $\alpha$-helical plug moiety. The cryo-EM structure reveals that the lateral gate and translocation pore of

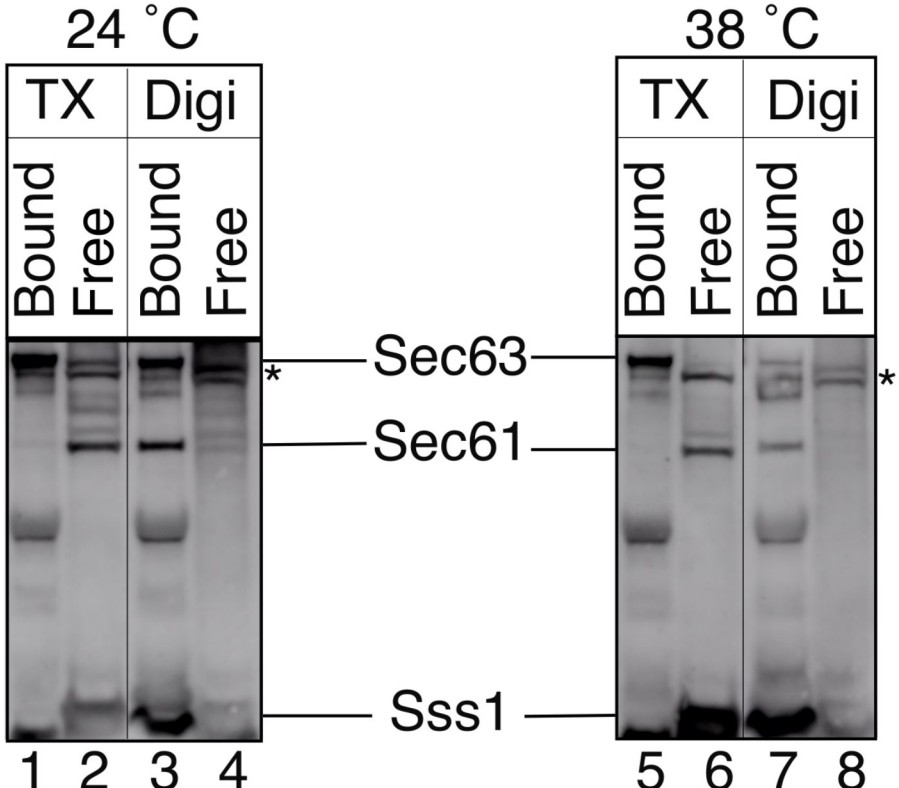

**Fig 7. The Sec complex remains stable in the absence of Sbh1.** Microsomes derived from Δsbh1 Δsbh2 yeast grown at permissive (24 ˚C) and restrictive (38 ˚C) temperature were solubilized in 3% (w/v) digitonin (Digi) or 1% (w/v) Triton X-100 (TX). Insoluble components were removed by centrifugation and cleared lysates were incubated with Concanavalin A-Sepharose. Unbound complexes (Free) and Concanavalin A-bound proteins (Bound) were analyzed by SDS-PAGE and immunoblotting with specific antibodies against Sec63, Sec61, and Sss1. The asterisk marks a doublet cross-reacting with the Sec63 antibody.

Sec61 adopt a wide-open conformation in the Sec61-Sec63 complex [5, 6] that likely keeps the channel activated for substrate engagement. Whereas the plug region is apparently not of vital importance for yeast cells, plug deletion mutants showed an effect on signal sequence orientation of diagnostic signal-anchor proteins, a minor defect in cotranslational, and a significant deficiency in posttranslational translocation [57]. Therefore we focused on these regions to examine the local conformational flexibility of Sec61 in the absence of Sec63.

**Resilience of the pore-ring and lateral gate.** The pore-ring of yeast Sec61 consists of the six residues V82(TM2), I86(TM2), I181(TM5), T185(TM5), M294(TM7) and M450(TM10) that are located at the N-termini of TM2, TM5, TM7 and TM10, respectively. To measure the conformational changes of the pore-ring, we defined two diagonal distance parameters, namely $d_{nTM2-nTM10}$ and $d_{nTM5-nTM7}$ (S10(A) Fig). $d_{nTM2-nTM10}$ is defined as the distance between the COM of Cα atoms of the N-terminal helical turns of TM2 and TM10. Similarly, $d_{nTM5-nTM7}$ describes the analogous distance between the N-terminal helical turns of TM5 and TM7. The probability distributions of $d_{nTM2-nTM10}$ during the last 50 ns of the MD simulations (Fig 8) indicate that the distance shrinks when Sec63 is absent (hypergeometric $p$-value = 0.004, threshold: average $d_{nTM2-nTM10} < 1.8$ nm). The average values with standard error (over five simulations) are 1.71±0.1 nm and 1.89±0.03 nm in the free and Sec63-bound states, respectively. The values are reasonably close to the available experimental cryo-EM structures.

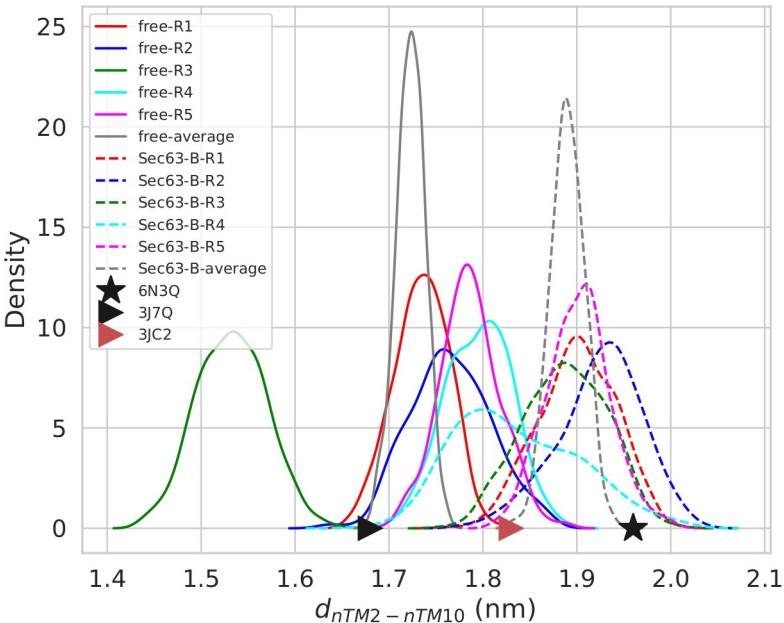

**Fig 8. The probability distributions of $d_{nTM2-nTM10}$ values obtained from the last 50 ns of five MD simulation replicas (Ri, i = replica number).** Solid and dashed lines represent 'free' (free; without Sec63) and 'Sec63-B' (Sec63-bound) states, respectively. Black star, black triangular and red triangular symbols represent the values in the experimental cryo-EM structures of Sec complex, idle-state ribosome-Sec61 complex, and open-state ribosome-Sec61 complex, respectively.

There, the distance is about 0.25 nm shorter in the idle-state ribosome-Sec61 complex (1.68 nm) than in the Sec complex (1.96 nm) and in the open-state ribosome-Sec61 (1.83 nm) complex, respectively. S11 Fig shows the evolution of $d_{nTM2-nTM10}$ values along with simulation time.

To inspect potential rearrangements of the narrow pore, we measured the angular shift of TM2 ($\theta_{TM2}$) with respect to the parallel vector of the lipid-bilayer plane (see S10(B) Fig). S12 Fig illustrates that the angle differs by about 6° (on average) between the two states (Sec63-bound and free). The average values (over five simulations and standard error) are 109.5±5.9° and 116.6±2.7° in free and Sec63-bound states, respectively. In spite of the insignificant change of $\theta_{TM2}$ (hypergeometric $p$-value = 0.1, threshold: average value < 115°) between the two states, $\theta_{TM2}$ and $d_{nTM2-nTM10}$ are linearly dependent (Pearson correlation coefficient = 0.82; see S13(A) Fig). Also, this angle is about 5° wider in the experimental cryo-EM structures of the open-state ribosome-Sec61 and Sec complexes compared to the idle-state ribosome-Sec61 complex. It affirms that TM2, a part of the lateral gate, is associated with the channel opening triggered by the association of Sec61 with Sec63 even though TM2 does not directly interact with Sec63. Residues of TM4 of Sec61, however, which is in direct contact with TM2, are located within 1.0 nm interaction distance of Sec63 in the cryo-EM structure [5, 6]. To understand how Sec63 affects the pore opening, we monitored the distance between the middle helical turns of TM2 and TM4 ($d_{mTM2-mTM4}$). $d_{mTM2-mTM4}$ is about 0.2 nm wider in the cryo-EM structure of the Sec complex than in that of the ribosome-Sec61 complex (see S14 Fig). Similarly, simulation results reproduced the decreasing trend of $d_{mTM2-mTM4}$ when Sec61 is not bound to Sec63 (hypergeometric $p$-value = 0.08, threshold: average value < 1.9 nm). $d_{mTM2-mTM4}$ and $\theta_{TM2}$ are also linearly dependent (see S13(B) Fig). The results suggest that shifting of TM4 toward TM2 causes the pore closing in the free state.

TM4 of Sec61 and TM3 of Sec63 both interact with TM1 of Sec61 and the TM helix of Sbh1. The position of the C-terminus of TM4 relative to the N-terminus of TM1 is described by $\theta_{cTM4nTM1}$ (S10(B) Fig). The values are 64.4˚, 85.5˚ and 128.0˚ in the cryo-EM structures of the Sec complex, the idle state of the ribosome-Sec61 complex, and the open state of the ribosome-Sec61 complex, respectively. The value is smaller in the Sec complex than in the other ribosome associated structures. S15(A) Fig shows that the same trend is also followed in the Sec63-bound and free simulations. The angle is about 6˚ (on average) wider in the free state compared to the Sec63-bound state (hypergeometric $p$-value = 0.02, threshold: average value $\geq$ 75˚).

In contrast to $d_{nTM2-nTM10}$, the other diagonal distance of the channel pore $d_{nTM5-nTM7}$ is about 0.4 nm larger in two out of five free simulations than in the Sec63-bound simulations (hypergeometric $p$-value = 0.2, threshold: average value > 1.8 nm) (see Fig 9). In those two free simulations, $d_{nTM5-nTM7}$ values increased prominently after 900 ns (S16 Fig). Surprisingly, the values obtained from the two free simulations are close to the ones from cryo-EM structure of the open-state ribosome-Sec61 complex (1.85 nm), whereas the values are 1.58 nm and 1.44 nm in the Sec complex and the idle-state ribosome-Sec61 complex, respectively. In Fig 8 ($d_{nTM2-nTM10}$), the distance in the Sec complex (black star) is larger than in the two ribosome-bound complexes. In contrast, $d_{nTM5-nTM7}$ (see Fig 9) for the Sec complex is in between that of the ribosome-bound complexes. Still, the result indicates that $d_{nTM5-nTM7}$ is also affected by Sec63.

The above analysis reveals that the conformation of the pore ring region is different in Sec63-bound and free states. In the absence of Sec63, $d_{nTM2-nTM10}$ shrinks and $d_{nTM5-nTM7}$ expands. If Sec61 is not bound to Sec63, the orientation of its TM4 helix is altered, which governs the movement of the C-terminus of TM2 towards the N-terminus of TM10 leading to a shortening of $d_{nTM2-nTM10}$. The cause behind the changed value of $d_{nTM5-nTM7}$ in the free state

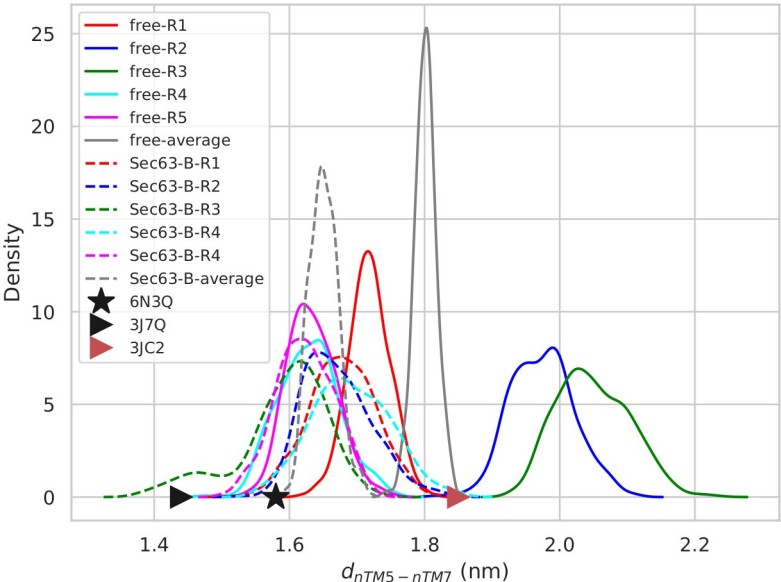

**Fig 9. The probability distributions of $d_{nTM5-nTM7}$ values obtained from the last 50 ns of five MD simulation replicas (Ri, i = replica number).** Solid and dashed lines represent 'free' (free; without Sec63) and 'Sec63-B' (Sec63-bound) states, respectively. Black star, black triangular and red triangular symbols represent the values in the experimental cryo-EM structures of Sec complex, idle-state ribosome-Sec61 complex and open-state ribosome-Sec61 complex, respectively.

simulation is not clear from the simulations. It is possible that the change in the $d_{nTM2-nTM10}$ leads to the alteration of $d_{nTM5-nTM7}$.

A structural superposition of the open-state cryo-EM structures (S17 Fig) suggests that a tilting motion of TM7 is involved in the lateral gate closing mechanism. S18 Fig shows that the angle $\theta_{TM7}$ between the TM7 helical axis and the parallel axis to the lipid bilayer plane changed towards the value of the ribosome-bound idle-state cryo-EM structure in all Sec63-bound and free simulations. The value of $\theta_{TM7}$ exhibited a rapid decrease in the free simulations (around 50 ns), unlike the Sec63-bound simulations (around 400 ns) (see S18(C) Fig). The result suggests that the parts of Sec63 included in our simulations have a negligible influence on the movement of TM7.

Additionally, we measured the distance between the N-termini of TM2 and TM7 ($d_{nTM2-nTM7}$) and the radius of the pore-ring region. S19 and S20 Figs show that $d_{nTM2-nTM7}$ and the radius of the pore-ring region shrink by $\sim$ 0.4 nm and $\sim$ 0.2 nm (on average) with respect to the starting conformation in both states ('Sec63-bound' and 'free'), respectively. The results suggest that the parts of Sec63 included in our simulations are not responsible for the wider $d_{nTM2-nTM7}$ and pore-ring radius in the cryo-EM structure. Although the simulations cannot conclusively reveal the cause for the changed value of $d_{nTM2-nTM7}$ and the pore-ring radius in both states simulations. Yet, it is plausible that the movement of TM7 is responsible for this. The radius of the pore-ring was calculated using a curve fitting approach. The C$\alpha$ atoms of the N-termini of TM2, TM5, TM7 TM10 and their contact residues (<0.5 nm distance) were projected on the plane of the lipid bilayer (XY-plane). Then, the equation of a circle was fitted through these projected points (S20 Fig).

**Orientation of the plug region.** The EM-structure (6ND1) [6] demonstrates that the plug region (Y64-A71) adopts an alpha-helical structure. The plug orientation in the MD simulations was characterized by the angle between the plug helical axis and the normal vector to the lipid bilayer (S10(C) Fig). The probability distributions of $\theta_{plug}$ (Fig 10), which is calculated from the final 50 ns of the trajectories, demonstrate that the angle increased by about 20° in two free simulations (hypergeometric $p$-value = 0.2, threshold: average value > 60°) in agreement with the trend seen in the cryo-EM structures. The values are 36.5° and 62.4° in the cryo-EM structures of Sec complex and idle-state ribosome-Sec61 complex, respectively. The average values with standard error (over five simulations) of $\theta_{plug}$ are 39.7±6.8° and 51.2±14.5° in the Sec63-bound and free states, respectively. The time-dependent $\theta_{plug}$ profiles (S21 Fig) show that the plug helix tends to fluctuate more in the free state than in the Sec63-bound state. The result suggests that Sec63 holds the plug in a particular orientation in the Sec complex. There is also a linear association between $\theta_{plug}$ and $d_{nTM2-nTM10}$ (see S22 Fig).

Our extensive MD simulations revealed that TM2, TM4, TM7 and the plug of Sec61 adopt different conformations in both Sec63-bound and free states. We wondered whether these differences would result in consequences for protein function and how this could be tested by computational means. It was previously shown that mutant alleles of SEC63 influence translocation of SRP-independent and SRP-dependent substrates in a different manner [16, 17]. Hence, we asked whether the Sec61 regions showing conformational shifts upon binding to Sec63 (TM2, TM4, TM7 and the plug) are in contact with substrate peptides during translocation. As a suitable technique to answer this question, we selected the molecular docking tool ADCP for placing substrate peptides into the Sec61 pore. As explained in the methods section, docking of full-length peptides or even of full-length signal sequences is computationally not feasible. However, we showed that docking of the hydrophobic segment of the CPY signal sequence gave very similar results to docking runs of the full-length CPY signal sequence (see Materials and methods). Hence, we focused the following docking study on the central hydrophobic parts of the signal sequences of SRP-dependent vs. SRP-independent substrates.

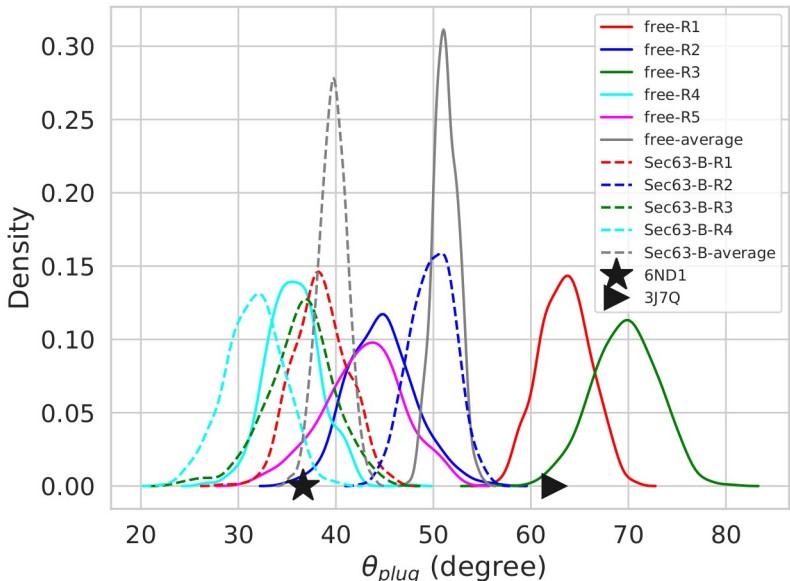

**Fig 10. The probability distributions of $\theta_{plug}$ values obtained from the last 50 ns of five MD simulation replicas (Ri, i = replica number).** Solid and dashed lines represent simulations of the 'free' (free) and 'Sec63-bound' (Sec63-B) states, respectively. Black star and triangular symbols represent values in the experimental cryo-EM structures of Sec complex and ribosome-Sec61 complex, respectively. The plug region is missing in the cryo-EM structure of the open-state ribosome-Sec61 complex (PDB ID: 3JC2).

## Interaction of the hydrophobic cores of signal sequences with the lateral gate region of Sec61

For the shorter SRP-independent signal sequences, the ADCP program found essentially two clusters of favorable docking poses, see Fig 11. In one of them, the hydrophobic core of the signal sequence aligns parallel to TM2 in the lateral gate. The poses of the other cluster are also placed inside the channel pore near the plug region, almost perpendicular to TM7. For SRP-dependent signal anchors, the docking poses are also located in the vicinity of the lateral gate, but are spread over a much longer volume. Furthermore, the docking results suggest that the hydrophobic cores of signal sequences like to form hydrogen bonds with the C-terminus of TM2 (I91-T98, cTM2) and the N-terminus of TM7 (T291-A298, nTM7) that form the lateral gate of the channel. Fig 11 shows the percentage of the total final docking poses where at least one residue of the hydrophobic core of the signal sequence is located near ($\leq$ 0.3 nm residue-residue distance) the cTM2 and nTM7 regions of Sec61. This percentage value reflects the possibility of forming a hydrogen-bond between the hydrophobic core and the lateral gate. Also it illustrates the possibility of occupying the volume between TM2 and TM7 helices (the lateral gate). Only 16.67%, 13.33%, 10.00% and 13.33% of the final docking poses of CPY, Gas1, prepro $\alpha$-factor and PDI, respectively, satisfy the above mentioned distance criterion. In contrast, more than 45.0% of the final docking poses of Dap2 (53.33%) and Pho8 (50.00%) show this behavior. Therefore, the docking results suggest that the hydrophobic cores of SRP-dependent signal anchors have a stronger tendency to interact with the lateral gate region than SRP-independent signal sequences (hypergeometric $p$-value = 7.87 x $10^{-8}$). $P$-values were calculated using a population size of 180 conformations (6 signal sequences x 30 final docking poses), a sample size of 60 conformations (2 SRP-dependent signal sequences x 30 final docking poses) and contact with the lateral gate region as threshold. The docking results confirm that the regions of the Sec61 pore that show conformational

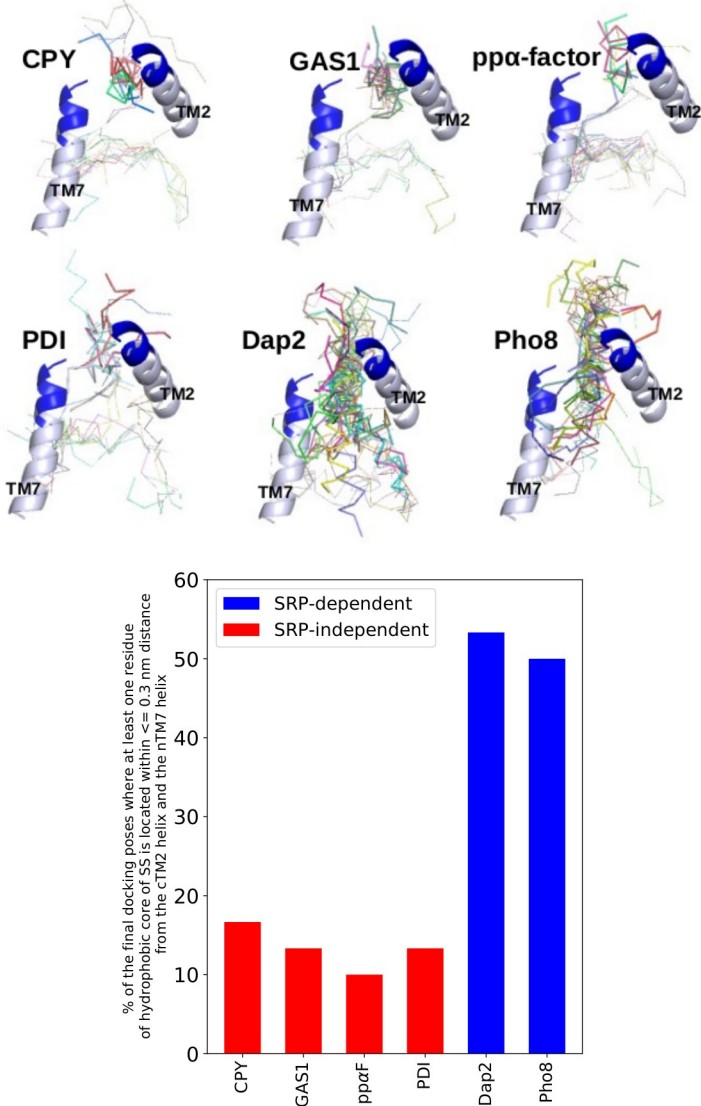

**Fig 11.** Upper-panel: All final docking poses are represented as line models. The final docking poses which satisfy the distance criterion are shown with thicker lines. TM2 and TM7 of Sec61 are shown in cartoon model. The blue parts of TM2 and TM7 represent the C-terminus of TM2 and the N-terminus of TM7. Instead of the whole Sec61 protein only TM2 and TM7 of Sec61 are shown for clarity. Lower-panel: The percentage of the total final docking poses where at least one residue of the hydrophobic core is located near ($\leq$ 0.3 nm residue-residue distance) the C-terminus of TM2 and the N-terminus of TM7. Basically it represents the percentage of conformations which occupy the volume between cTM2 and nTM7 (lateral gate). The pp*alpha*-factor is abbreviated as pp$\alpha$F.

shifts between Sec63-bound and free (without Sec63) states are involved in binding of Sec61 substrate peptides.

## Discussion

The cryo-EM structures of the Sec61 complex bound to Sec63 revealed that the conformation of the Sec61 channel is influenced by the Sec62-Sec63 complex, although it remained unclear how the interactions between Sec61 and Sec63 precisely contribute to the wide-opening of the Sec61 channel. Here, we characterized the role of Sec63 in maintaining the Sec61 channel in

an open conformation using MD simulations. MD results suggest that the conformational dynamics of the lateral-gate, pore-ring and plug regions change if Sec63 is not attached to Sec61. We will now discuss these conformational effects one by one and connect them to existing experimental and computational evidence.

To interpret the simulation results, we need to mention again that the J-domain and 63 residues from the C-terminus of Sec63 are missing in the simulations, due to the lack of experimental structural information. Our simulation results (Fig 5) show that the missing parts (mostly) do not affect the extensive contacts between Sec63 and Sec61. The interactions between Sec61$\alpha$ and Sec63 remained stable throughout the 1 $\mu$s long simulations. The interaction of Sec63 with the $\beta$-subunit of Sec61 complex (Sbh1) (Fig 5C) at IS2 also remained stable during the simulations. Nevertheless, our co-precipitation experiments with $\Delta$sbh1 Sec complex (Fig 7) revealed that Sbh1 does not play an essential role for the stability of the Sec complex. Partial unfolding of Sec61 (P200-E212) at the lumenal interaction site (IS3) was observed in the absence of Sec63 (Fig 6) in the simulations. P200-E212:Sec61 adopted a $\beta$-sheet conformation with S205-G207:Sec63. Wu et al. [6] showed that point-mutations of conserved residues at the N-terminus of Sec63 (Y5A, Y7A and D8A) in the IS3 region do not affect the function of Sec63. However, these residues of Sec63 are not interacting residues (located within $\leq$ 0.5 nm of Sec61) at IS3 whereas the S205-G207 residues of Sec63 make contact with Sec61 at IS3. Therefore, we suggest testing the effect of point-mutations of S205-G207:Sec63 on the integrity and function of the Sec complex. Further, it will be interesting to explore the effect of mutating residues P200-E212:Sec61 on protein translocation.

Judged by the distance between the N-termini of TM2 (S83-I86) and TM10 (S447-M450) ($d_{nTM2-nTM10}$), the pore remains wider (ca. 0.2 nm) in the presence of Sec63, in agreement with the experimental structures (Fig 8). Our thorough analysis of the structural dynamics suggests that a motion of TM2 towards TM10 is involved in the pore closing mechanism. Also, the orientation of TM2 was found to be associated with that of TM4. This observation is consistent with the suggestion of Wu et al. [6] that the lateral gate opening is caused by TM4. Furthermore, our analysis suggests that Sec63 regulates the pore conformation via interacting with TM1 of Sec61 (S15 Fig). The interaction between Sec63 and Sbh1 may also be associated with the pore closing mechanism, because the distance between the middle helical-ring of TM4 of Sec61 and the TM of Sbh1 is about 0.2 nm smaller in the ribosome-Sec61 complex (I161-L164:TM4-Sec61 and 39-42:Sbh1; PDB ID: 3JC2 [19]) than in the Sec complex (S161-V164:TM4-Sec61 and L60-G63:Sbh1; PDB ID: 6ND1 [6]). The movement of the TM of Sbh1 towards TM4, however, was not captured in the free simulations, possibly due to the limited time-scale of the simulations. Besides, we observed that the distance between the N-termini of TM5 and TM7 ($d_{nTM5-nTM7}$) was different in the open and closed states of the pore (Fig 9).

Our results suggest conformational changes in the pore region of Sec61 in the Sec63-bound state compared to the free state. The six apolar residues of the pore ring, V82, I86, I181, T185, M294, and M450, are located at the N-termini of the TM2, TM5, TM7 and TM10 helices in the pore region. Junne et al. [10] demonstrated the importance of these residues for protein translocation using mutagenesis. The mutations not only changed the physicochemical properties of the pore, but may have also changed the conformation of the pore region [58]. Therefore, we speculate that the conformation of the pore ring region may play an important role for protein translocation.

It has been shown by cysteine cross-linking that the plug movement is triggered by polypeptide translocation through SecY [59, 60]. In contrast to Wu et al. [6], the plug is unresolved in the cryo-EM structure of the yeast Sec complex by Itskanov and Park [5]. Molecular dynamics simulations suggested that the plug position is connected to the opening of the lateral gate

[26]. Our study shows that the orientation of the plug is influenced by the presence of Sec63, in agreement with experimental structures (Fig 10), suggesting that the orientation of the plug is indeed influenced by the altered structures of the pore ring and lateral gate triggered by Sec63.

Recently, combination of biochemical experiments and MD simulations revealed a signal sequence-induced movement of TM7 of bacterial SecY [61]. Mutagenesis studies have demonstrated the importance of TM7 for protein translocation through the eukaryotic Sec61 channel as well [10, 62]. We noticed that the angle between the TM7 helical axis and the normal to the lipid-bilayer is about 20° wider in the cryo-EM structure of the opened ribosome-Sec61 complex than in the closed conformation. This angle is also about 10° wider in the Sec complex than in the opened ribosome-Sec61 complex (S18 Fig). In our simulations, TM7 gradually adopted the closed gate conformation in both conditions (Sec63-bound and free) (S18 Fig). The closing process, however, was slower in the Sec63-bound state (S18(C) Fig). The parts of Sec63 included in our simulations have a negligible influence on the movement of TM7.

Our simulation results suggest that the conformation of the pore-ring region is associated with Sec63. The results illustrate that Sec63 keeps $d_{nTM2-nTM10}$ wider/longer by interacting with TM4 of Sec61. The change in $d_{nTM2-nTM10}$ may lead to the observed alteration of the pore region conformation. The distance between the N-termini of TM2 and TM7 (S19 Fig) and the radius of the pore-ring (S20 Fig) adopted the closed gate conformation in both conditions (Sec63-bound and free). These measurements are done with respect to TM7, which is a part of the lateral-gate. Furthermore, TM7 had a similar orientation in the simulations of both states (S18 Fig).

However, the simulations cannot answer what causes the TM7 movement in the simulations of the Sec63-bound state. We assumed that the parts of Sec63 included in our simulations have a negligible influence on the movement of TM7. Recently published cryo-EM structures [63, 64] shed light on the orientation of TM7 in the Sec complex. They revealed that interactions between TMs of Sec62 with TM7-Sec61 keep the lateral-gate open in the Sec complex. Besides, Itskanov et al. [63] revealed that the Sec61 channel remains inactive without Sec62. This cryo-EM structure of Sec62 was not available when we conducted our simulations. Hence, the TMs of Sec62 are not present in our Sec63-bound simulations. Now, based on our MD simulations and the new cryo-EM structures, it has become clear that the conformation of the lateral gate and the pore-region are associated with both Sec62 and Sec63.

The ion-conducting characteristic of the Sec61 channel the yeast is not well studied yet. The radius profiling (S23 Fig) shows that the radius of the narrowest point of the pore channel is <1.5 Å in the resting condition or closed-state (without substrates). Further, the channel is blocked by a small $\alpha$-helical plug from the luminal side. In this static resting conformation, ion movement along the channel pore appears not feasible. However, Erdmann et al. [65] demonstrated that the mammalian (*C. familiaris*) and the yeast (*S. cerevisiae*) Sec61 channels exhibit almost identical marginal anion selectivity characteristics. In higher eukaryotes (mammalian cells), the heterotrimeric Sec61 complex in the ER membrane provides an aqueous pathway for calcium leakage from the ER to the cytosol [66]. In yeast, the Sec61 channel may serve as a conduit for small, uncharged peptides like glutathione and the bivalent manganese cation [67, 68].

Alanine-scanning and *in vivo* site-directed photocross-linking experiments demonstrated that mutations in TM2 and TM7 lead to defects in translocation and membrane insertion of proteins [69]. Molecular dynamics simulations by Zhang and Miller suggested that the plug remains positioned either toward the lateral gate for a hydrophilic substrate or toward the inner side of the pore for a hydrophobic substrate [27]. Here, we performed molecular docking of the hydrophobic core of signal sequences and signal anchors to the cryo-EM structure of

Sec61 in its conformation in the heptameric Sec complex. The aim of this was to test whether those Sec61 structural elements affected by Sec63 are indeed related to substrate translocation of membrane insertion. Docking showed that the hydrophobic core of signal anchors of SRP-dependent substrates more likely occupies the volume between the C-terminus of TM2 and the N-terminus of TM7 than the hydrophobic core of signal sequences of SRP-independent substrates, suggesting that the translocation process depends on the interaction of targeting sequences with the lateral gate in agreement with other studies [69]. The hydrophobic core of SRP-dependent signal anchors is longer and more hydrophobic compared to SRP-independent signal sequences. To examine the effect of the hydrophobicity of the hydrophobic core on the interaction with the lateral gate, we then docked a hydrophobic CPY-variant (S24 Fig) and found that the hydrophobic core of the CPY-variant is more likely to interact with the lateral gate than the hydrophobic core of the original CPY signal sequence. Moreover, our MD simulation results demonstrate that the orientation of the TM2 and TM7 helices are dependent on the interaction of Sec61 and Sec63, suggesting that the absence of Sec63 alters the orientation of these helices, which may affect protein translocation differentially depending on targeting sequence hydrophobicity. Mutation at the N-terminus of Sec62 (which interacts with the C-terminus of Sec63 [15]) and the C-terminus of Sec63 impaired the translocation of SRP-independent substrates, whereas SRP-dependent ones were unaffected [17]. Our docking results confirm the findings of previous studies [24, 25, 27] that the hydrophobicity of the hydrophobic core plays an important role for protein translocation. Due to the narrow size of the pore, substrates cannot be docked inside the Sec61 channel in the closed-state conformation. Therefore, how these substrates interact with the lateral-gate in the closed-state of Sec61 cannot be investigated by molecular docking.

## Conclusion

In summary, the results presented here provide novel insight into the influence of Sec63 on the conformational dynamics of the Sec61 channel of yeast as well as in eukaryotes. Our results describe how the interaction of Sec63 with Sec61 keeps the channel open. Despite its position at the interface beween Sec63 and Sec61, we found that Sbh1 is not critical for the stability of the Sec complex. Furthermore, the molecular docking study sheds light on the putative interaction pattern of SRP-dependent signal anchors and SRP-independent signal sequences with the lateral gate of the Sec61 channel of the Sec complex. Therefore, our study provides an atomic view of Sec61 channel dynamics. It may stimulate further mutational studies to understand how the lateral gate residues influence SRP-dependent and SRP-independent substrate translocation in eukaryotes. Furthermore, it will be fascinating to see whether Sec63 on its own acts differently on the channel than the tetrameric Sec62-Sec63 complex.

## Supporting information

**S1 Fig. The probability distributions of $d_{nTM2-nTM10}$ values obtained from the (A) last 50 ns (hypergeometric p-value = 0.004, threshold: average $d\_nTm2-nTm10 < 1.8$ nm) (B) last 400 ns (hypergeometric p-value = 0.02, threshold: average $d\_nTm2-nTm10 < 1.8$ nm) of five MD simulation replicas (Ri, i = replica number).** Solid and dashed lines represent 'free' (free; without Sec63) and 'Sec63-B' (Sec63-bound) states, respectively. Black star, black triangular and red triangular symbols represent the values in the experimental cryo-EM structures of Sec complex, idle-state ribosome-Sec61 complex, and open-state ribosome-Sec61 complex, respectively.
(TIF)

**S2 Fig. The probability distributions of $d_{nTM5-nTM7}$ values obtained from the (A) last 50 ns (hypergeometric $p$-value = 0.2, threshold: average value $>$ 1.8 nm) (B) last 400 ns (hypergeometric $p$-value = 0.2, threshold: average value $>$ 1.8 nm) of five MD simulation replicas (Ri, i = replica number).** Solid and dashed lines represent 'free' (free; without Sec63) and 'Sec63-B' (Sec63-bound) states, respectively. Black star, black triangular and red triangular symbols represent the values in the experimental cryo-EM structures of Sec complex, idle-state ribosome-Sec61 complex and open-state ribosome-Sec61 complex, respectively.
(TIF)

**S3 Fig. The probability distributions of $\theta_{plug}$ values obtained from the (A) last 50 ns (hypergeometric $p$-value = 0.2, threshold: average value $>$ 60˚) (B) last 400 ns (hypergeometric $p$-value = 0.2, threshold: average value $>$ 55˚) of five MD simulation replicas (Ri, i = replica number).** Solid and dashed lines represent simulations of the 'free' (free) and 'Sec63-bound' (Sec63-B) states, respectively. Black star and triangular symbols represent values in the experimental cryo-EM structures of Sec complex and ribosome-Sec61 complex, respectively. The plug region is missing in the cryo-EM structure of the open-state ribosome-Sec61 complex (PDB-ID:3JC2).
(TIF)

**S4 Fig. Heatmap represents the RMSD values between the final docking poses of the Sec61-CPY complexes (CPY signal sequences) and the Sec61-Hydrophobic region of CPY signal sequence complexes.** The bottom panel shows the conformation with lowest RMSD value. Sec61 (white), hydrophobic region of CPY (blue) and other regions of CPY (red) are shown in white, blue, and red, respectively.
(TIF)

**S5 Fig. RMSF analysis of Sec61 residues during five independent MD simulations (Ri, i = replica number).** (A) and (B) represent RMSF values in the individual simulations at Sec63-bound and free state, respectively. (C) shows averages of the five simulations. IS3 and IS1 regions are magnified. IS3 (hypergeometric $p$-value = 0.023, threshold: RMSF of Gly206 $>$ 0.55 nm) and IS1 (hypergeometric $p$-value = 0.004, threshold: RMSF of Arg275 $>$ 0.5 nm) regions are significantly more flexible in free state simulations compared to the Sec63-bound state.
(TIF)

**S6 Fig. DSSP profile of Sec61 at the interaction site 3 (IS1) (R264-P282) from five independent MD simulations (Ri, i = replica number).** (A) Sec63-bound state (B) Free state (without Sec63).
(TIF)

**S7 Fig. DSSP profile of Sec61 at the interaction site 1 (IS3) (P200-E212) from five independent MD simulations (Ri, i = replica number).** (A) Sec63-bound state (B) Free state (without Sec63).
(TIF)

**S8 Fig. Probability distribution of the percentage of residues of Sec61 at IS1 (R264-P282) adopting $\beta$-sheet conformation during the last 50 ns of MD simulations.** 'Sec63-B' and 'free' denote Sec63-bound and free states, respectively. The value is 65% in both cryo-EM structures of the Sec complex and of the ribosome associated Sec61 complex.
(TIF)

**S9 Fig. The conformational changes at interaction sites are illustrated by structural superimposition of the final conformation (at 1μs) of Sec61 in the Sec63-bound (cyan) and free (blue) states (replica 1).**
(TIF)

**S10 Fig. Definition of contact distances and angles to characterize the local conformation of Sec61.** (A) $d_{nTM2-nTM10}$, $d_{nTM5\_nTM7}$ and $d_{nTM2\_nTM7}$ are the distances between the center-of-mass of the N-terminal helical turns of TM2 (S83-I86) & TM10 (S447-M450), TM5 (I181-F184) & TM7 (P292-L295) and TM2 (S83-I86) & TM7 (P292-L295), respectively. (B) The orientation of TM2 is defined by the angle between the TM2 helical axis (black) and the vector parallel to the lipid layer oriented along the x-axis (red), ($\theta_{TM2}$). The distance between TM2 and TM4 helices is measured by the distance between COM of Cα atoms of the middle helical turn of TM2 (S89-F92) and TM4 (M158-S161). The orientation of the C-terminus of TM4 (C168-L171) with respect to the N-terminus of TM1 (N30-L33) is described by the angle $\theta_{cTM4nTM1}$. (C) The orientation of the plug helix with respect to the lipid bilayer is measured by the angle between the plug helical axis (black) and the vector normal to the lipid layer (red), $\theta_{Plug}$. In superimposed structures blue, and cyan colors represent free and Sec63-bound states, respectively.
(TIF)

**S11 Fig. Time-dependent $d_{nTM2-nTM10}$ profiles during the five MD simulation replicas (Ri, i = replica number).** 'Sec63-B' and 'free' denote Sec63-bound and free states. The right panel also includes Sec63-B-avg from the left panel to illustrate the deviation between Sec63-B-avg and free-avg. (see also S12, S14, S16 and S18 Figs). (A) Sec63-bound state (B) Free state (without Sec63).
(TIF)

**S12 Fig.** (A) Probability distributions of $\theta_{TM2}$ values obtained from the last 50 ns of five MD simulation replicas (Ri, i = replica number). Solid and dashed lines represent free (free) and Sec63-bound (Sec63-B) states, respectively. Black star, black triangular and red triangular symbols represent the corresponding values in the experimental cryo-EM structures of Sec complex, idle-state ribosome-Sec61 complex and open-state ribosome-Sec61 complex, respectively. Time-dependent $\theta_{TM2}$ profiles during the five MD simulation replicas (B) Sec63-bound state (C) Free state (without Sec63).
(TIF)

**S13 Fig. Dependence of (A) $\theta_{TM2}$ on $d_{nTM2-nTM10}$ (Pearson correlation coefficient 0.82) and (B) $\theta_{TM2}$ on $d_{mTM2-mTM4}$ (Pearson correlation coefficient 0.75), based on the final 50 ns of five independent MD simulations (Ri, i = replica number).**
(TIF)

**S14 Fig.** (A) Probability distributions of $d_{mTM2-mTM4}$ distances obtained from the last 50 ns of five MD simulation replicas (Ri, i = replica number). Solid and dashed lines represent free/ without Sec63 (free) and Sec63-bound (Sec63-B) states, respectively. Black star, black triangular and red triangular symbols represent the corresponding values in the experimental cryo-EM structures of Sec complex, idle-state ribosome-Sec61 complex and open-state ribosome-Sec61 complex respectively. Time-dependent $d_{mTM2-mTM4}$ profiles during the five MD simulation replicas (B) Sec63-bound state (C) Free state (without Sec63).
(TIF)

**S15 Fig.** (A) Probability distributions of $\theta_{cTM4nTM1}$ angles obtained from the final 50 ns of five independent MD simulation replicas (Ri, i = replica number). Solid and dashed lines represent

free (free) and Sec63-bound (Sec63-B) states, respectively. Black star and black triangular marks represent the respective values in the experimental cryo-EM structures of Sec and ribosome-Sec61 (closed state) complexes, (B) Dependence of $\theta_{cTM4nTM1}$ on $d_{mTM2-mTM4}$, during the final 50 ns of the five independent MD simulations (Pearson correlation coefficient -0.70). The values corresponding to the open-state ribosome-Sec61 complex are not included in the plots because their values differ largely. They are only given in the legend.
(TIF)

**S16 Fig. Time-dependent $d_{nTM5-nTM7}$ profiles during the five MD simulation replicas (Ri, i = replica number).** (A) Sec63-bound state (B) Free state (without Sec63).
(TIF)

**S17 Fig. Superimposed cryo-EM structures of the Sec complex (gray; 6N3Q), idle-state ribosome-Sec61 complex (green; 3J7Q) and open-state ribosome-Sec61 complex (dark red; 3JC2).** The $\theta_{TM7}$ angle describes the orientation of the TM7 helix with respect to the plane of the membrane. The angle, $\theta_{TM7}$, is defined as the angle between the TM7 helical axis (black) and the vector parallel to the lipid layer oriented towards the x-axis (red).
(TIF)

**S18 Fig.** (A) Probability distributions of $\theta_{TM7}$ angles obtained from the final 50 ns of five independent MD simulation replicas (Ri, i = replica number). Solid and dashed lines represent free/without Sec63 (free) and Sec63-bound (Sec63-B) states, respectively. Black star, black triangular and red triangular symbolss represent the respective values in the experimental cryo-EM structures of Sec complex, idle-state ribosome-Sec61 complex and open-state ribosome-Sec61 complex, respectively. Time-dependent $\theta_{TM7}$ profiles during the five MD simulation replicas (B) Sec63-bound state (C) Free state (without Sec63).
(TIF)

**S19 Fig. The probability distributions of $d_{nTM2-nTM7}$ values obtained from the last 50 ns of five MD simulation replicas (Ri, i = replica number).** Solid and dashed lines represent simulations of the 'free' (free) and 'Sec63-bound' (Sec63-B) states, respectively. Black star and triangular symbols represent values in the experimental cryo-EM structures of Sec complex and ribosome-Sec61 complex, respectively.
(TIF)

**S20 Fig.** Upper panel: (A) The probability distributions for the radius of the pore-ring region obtained from the last 50 ns of five MD simulation replicas (Ri, i = replica number). Solid and dashed lines represent simulations of the 'free' (free) and 'Sec63-bound' (Sec63-B) states, respectively. Black star and triangular symbols represent values in the experimental cryo-EM structures of Sec complex and ribosome-Sec61 complex, respectively. Lower panel: (B) The radius of the pore-ring region was calculated using a curve fitting approach. The C$\alpha$ atoms of the N-termini of TM2, TM5, TM7, TM10 and their contact residues (<0.5 nm distance)(red color sphere) were projected on the lipid bilayer plane (X-Y plane). Then, the equation of a circle was fitted to these projected points.
(TIF)

**S21 Fig. Time-dependent $\theta_{plug}$ profiles during the five MD simulation replicas (Ri, i = replica number).** (A) Sec63-bound state (B) Free state (without Sec63).
(TIF)

**S22 Fig. Dependence of $\theta_{plug}$ on $d_{nTM2-nTM10}$, during the final 50 ns of the five independent MD simulations (Ri, i = replica number) (Pearson correlation coefficient 0.63).** The plug

region is missing in the cryo-EM structure of the open-state ribosome-Sec61 complex (PDB ID: 3JC2).
(TIF)

**S23 Fig. Radius profile of Sec61 in PDB ID 6ND1 obtained from the MOLEOnline webserver [70].**
(TIF)

**S24 Fig.** Left-panel: All final docking poses of the **CPY-variant** (LLTLLLCLLLL) [15] are represented by line model. The final docking poses that satisfy the distance criterion are shown with thicker lines. Sec61, the TM2 helix and the TM7 helix are shown in surface model and cartoon model, respectively. The blue parts of TM2 and TM7 helices represent the C-terminus of TM2 and the N-terminus of TM7. Right-panel: The percentage of the total final docking poses where at least one residue of the hydrophobic core is located near ($\leq 0.3$ nm residue-residue distance) the C-terminus of TM2 and the N-terminus of TM7.
(TIF)

**S1 Table. List of signal sequences used in this study.**
(XLSX)

## Acknowledgments

The authors thank PD Dr. Michael Hutter for a careful reading of the text.

## Author Contributions

**Conceptualization:** Pratiti Bhadra, Volkhard Helms.

**Data curation:** Pratiti Bhadra, Lalitha Yadhanapudi.

**Formal analysis:** Pratiti Bhadra, Lalitha Yadhanapudi.

**Funding acquisition:** Volkhard Helms.

**Investigation:** Pratiti Bhadra, Lalitha Yadhanapudi, Karin Römisch, Volkhard Helms.

**Methodology:** Pratiti Bhadra, Lalitha Yadhanapudi, Karin Römisch, Volkhard Helms.

**Project administration:** Pratiti Bhadra, Lalitha Yadhanapudi, Karin Römisch, Volkhard Helms.

**Resources:** Karin Römisch, Volkhard Helms.

**Software:** Pratiti Bhadra, Lalitha Yadhanapudi.

**Supervision:** Karin Römisch, Volkhard Helms.

**Validation:** Pratiti Bhadra, Lalitha Yadhanapudi.

**Visualization:** Pratiti Bhadra, Lalitha Yadhanapudi.

**Writing – original draft:** Pratiti Bhadra, Lalitha Yadhanapudi.

**Writing – review & editing:** Karin Römisch, Volkhard Helms.

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
