## [Decision Letter · Decision Letter 0]

22 Dec 2020

Dear Prof. Helms,

Thank you very much for submitting your manuscript "How does Sec63 affect the conformation of Sec61 in yeast?" for consideration at PLOS Computational Biology.

As with all papers reviewed by the journal, your manuscript was reviewed by members of the editorial board and by several independent reviewers. In light of the reviews (below this email), we would like to invite the resubmission of a significantly-revised version that takes into account the reviewers' comments.

We cannot make any decision about publication until we have seen the revised manuscript and your response to the reviewers' comments. Your revised manuscript is also likely to be sent to reviewers for further evaluation.

Sincerely,

Alexander MacKerell

Associate Editor

PLOS Computational Biology

Nir Ben-Tal

Deputy Editor

PLOS Computational Biology

Reviewer's Responses to Questions

**Comments to the Authors:**

Reviewer #1: This work investigates the effects of bounded Sec63 on the conformational dynamics of Sec61 channel through MD simulations. The translocation mechanism of Sec61/SecY complex is very complicated that involves the cooperation between partners in complex, and remains unclear so far. In this paper, the authors analysed the position and orientation changes of involved TM helices by the data comparison. The results will provide insights on understanding the translocation mechanism. The data of the simulations is reliable. However, the discussion about the mechanism how the interaction between Sec63 and TM4 effect the conformational change of pore ring, lateral gate is not convincing. Few issues need to be addressed before the manuscript can be accepted.

Major issues:

1. For the free simulations (Sec61 complex without Sec63), the authors simply removed Sec63 from the bound structure (PDB ID: 6N3Q). This will result in position and orientation changes of pore ring, lateral gate and plug. In the Fig. 1, the equilibrated structure is reached at around 600 ns of 1 μs simulations. But the data only the last 50 ns was used to be analysed.

2. In the Fig. 3, authors said "the contacts at IS2, IS3 and IS4 were well maintained, likely due to the regular secondary structure..." (see line 272-279), and the contact at IS1 was not stable that is related to the missing of some residues. The authors should show the region of bilayer in Fig. 3(a). The bilayer could make the contact more stable if the interaction site is located in the bilayer. The fluctuation of Fig. 3(b) likely due to the solvent too.

3. In the Fig. S4, the RMSF in the region near the IS1 and IS3 should be magnified to show the significant difference between the bounded and free state.

4. In the part of "Resilience of the pore-ring and lateral gate", the interaction between Sec63 and TM4 of Sec61 makes the θTM2 change. Then, it makes the dnTM2-nTM10 and dnTM2-nTM4 longer, but the dnTM5-nTM7 shorter. It seems disagree with the cryo-EM structure (line 350). The discussion of this part is not convincing. Does the Sec63 make the dnTM2-nTM7 longer? How the Sec63 makes pore and lateral gate wider is still not clear. Figures of the dnTM2-nTM7 and the diameter of the pore (like Fig. 6,7) should be helpful to current discussion.

Minor issues:

Two minor spelling mistakes: line 113: "Ssh1" should be "Sss1". line 655: "θTM7" should be "θplug".

Reviewer #2: In this paper, the authors have combined molecular modeling, i.e., docking

and MD simulations, with co-precipitation experiments on the Sec complex

especially focusing on Sec61 and Sec63 in yeast. The conformations and

their changes of Sec61 and the effect of the binding of Sec63 to Sec61 were

at the heart of this study. The simulations do show interesting changes in

the plug conformations depending on the presence/absence of Sec63

especially also of the lateral gate. Thus, the present study adds some nice

pieces to the understanding of the translocon and its accessory proteins.

- Add a brief summary of the experimental findings to the Author Summary

and the Conclusion section. A more detailed combination of theory and

experiment in the main text would be helpful for the reader.

- I would strongly suggest to include Fig S1 but also Fig S3 to be included

in the main text. The reader needs to get a chance to get acquainted with

the proteins and one cannot expect the reader to look into the SI for

such basic figures. Concerning Fig S1 I would suggest to make the

membrane a bit more visible.

- Could the channel in Sec61 be characterized by a radius profile? Is

anything known about its ion conductance?

Other than this, the manuscript is timely and interesting and certainly deserves publication.

**Have all data underlying the figures and results presented in the manuscript been provided?**

Reviewer #1: None

Reviewer #2: Yes

PLOS authors have the option to publish the peer review history of their article (what does this mean?). If published, this will include your full peer review and any attached files.

Reviewer #1: No

Reviewer #2: No
---

## [Decision Letter · Decision Letter 1]

5 Mar 2021

Dear Prof. Helms,

We are pleased to inform you that your manuscript 'How does Sec63 affect the conformation of Sec61 in yeast?' has been provisionally accepted for publication in PLOS Computational Biology.

Best regards,

Alexander MacKerell

Associate Editor

PLOS Computational Biology

Nir Ben-Tal

Deputy Editor

PLOS Computational Biology

Reviewer's Responses to Questions

**Comments to the Authors:**

Reviewer #1: In the revised manuscript, the authors carried out a comparison and analysis of distances and angles from last 50 ns and 400 ns respectively, and added distribution plots of dnTM2-nTM7 and pore-ring and a radius profile of channel. The complementary data and analysis makes conclusion more reliable and clear. The explanation for the mechanism of conformational changes is also reasonable. I would like to suggest publication.

Reviewer #2: The manuscript has been improved considerably and I now recommend publication as is.

**Have all data underlying the figures and results presented in the manuscript been provided?**

Reviewer #1: Yes

Reviewer #2: None

PLOS authors have the option to publish the peer review history of their article (what does this mean?). If published, this will include your full peer review and any attached files.

Reviewer #1: No

Reviewer #2: No

---

## [Editor Report · Acceptance letter]

25 Mar 2021

PCOMPBIOL-D-20-02132R1 

How does Sec63 affect the conformation of Sec61 in yeast?

Dear Dr Helms,

I am pleased to inform you that your manuscript has been formally accepted for publication in PLOS Computational Biology. Your manuscript is now with our production department and you will be notified of the publication date in due course.

With kind regards,

Katalin Szabo
